# Role of Ferroptosis in Non-Alcoholic Fatty Liver Disease and Its Implications for Therapeutic Strategies

**DOI:** 10.3390/biomedicines9111660

**Published:** 2021-11-10

**Authors:** Han Zhang, Enxiang Zhang, Hongbo Hu

**Affiliations:** 1College of Food Science and Nutritional Engineering, China Agricultural University, Beijing 100080, China; S20193060988@cau.edu.cn; 2Key Laboratory of Growth Regulation and Transformation Research of Zhejiang Province, School of Life Sciences, Westlake Institute for Advanced Study, Westlake University, Hangzhou 310024, China

**Keywords:** ferroptosis, NAFLD, lipid peroxidation, iron metabolism

## Abstract

Non-alcoholic fatty liver disease (NAFLD) has become the chronic liver disease with the highest incidence throughout the world, but its pathogenesis has not been fully elucidated. Ferroptosis is a novel form of programmed cell death caused by iron-dependent lipid peroxidation. Abnormal iron metabolism, lipid peroxidation, and accumulation of polyunsaturated fatty acid phospholipids (PUFA-PLs) can all trigger ferroptosis. Emerging evidence indicates that ferroptosis plays a critical role in the pathological progression of NAFLD. Because the liver is the main organ for iron storage and lipid metabolism, ferroptosis is an ideal target for liver diseases. Inhibiting ferroptosis may become a new therapeutic strategy for the treatment of NAFLD. In this article, we describe the role of ferroptosis in the progression of NAFLD and its related mechanisms. This review will highlight further directions for the treatment of NAFLD and the selection of corresponding drugs that target ferroptosis.

## 1. Introduction

Non-alcoholic fatty liver disease (NAFLD) is a chronic progressive liver disease with steatosis as the main pathological feature, including simple fatty liver degeneration, non-alcoholic steatohepatitis (NASH). It may develop into cirrhosis and liver cancer [1,2]. NAFLD is the most common chronic liver disease in the world today, and its incidence in the Euro-American region has reached more than 20% [3]. There are similarities and differences in the epidemiology of NAFLD in different regions of the world [4]. Metabolic syndrome is the basis of NAFLD. Modern research demonstrates that NAFLD is related to many factors, such as inflammation, oxidative stress, insulin resistance, lipid metabolism disorders, obesity, and endoplasmic reticulum stress [5]. Unfortunately, due to the unclear pathogenesis of NAFLD, there is currently no satisfactory strategy for treatment. As a new form of programmed cell death, the biochemical, morphological, and genetic characteristics of ferroptosis are different from other typical cell deaths (such as apoptosis, necrosis, pyrolysis, and autophagy-dependent cell death) [6,7,8,9]. Ferroptosis has been reported to play an important role in many human diseases. For example, ischemic organ injury (IOI) is accompanied by massive cell death in the affected organ [10], and inhibition of ferroptosis can protect IOI in mouse liver, kidney, brain, and heart models [11,12,13,14,15]. In the process of aging and degenerative diseases, brain iron levels will inevitably rise [16,17]. The use of ferroptosis inhibitors also has a protective effect on diseases such as Parkinson’s disease, Huntington’s disease, and Alzheimer’s disease [18,19,20,21]. Moreover, the important role of ferroptosis in cancer development and treatment is emerging. Studies have shown that many types of therapy-resistant cancer cells are more susceptible to ferroptosis [22,23,24], which indicates that induction of ferroptosis may be a promising cancer treatment that can overcome the drug resistance of tumor cells [25]. In particular, more and more evidence shows that ferroptosis plays a key role in the pathological process of liver diseases, including NAFLD [26,27]. It is currently believed that iron overload may be involved in the NAFLD process, and the ferroptosis of liver cells and intrahepatic macrophages may lead to the development of the simple fatty liver degeneration of NASH [28]. Therefore, inhibiting ferroptosis is likely to become a new treatment strategy for NAFLD in the future.

In this review, we briefly summarize the regulatory mechanism of ferroptosis and discuss how ferroptosis is involved in the development of NAFLD. We also discuss the latest research progress in drugs that target ferroptosis to treat NAFLD. Moreover, we propose potential therapeutic strategies involving ferroptosis for the treatment of NAFLD as well as current problems and the direction of future research in the corresponding fields. This will provide new opportunities for the prevention and treatment of NAFLD.

## 2. Ferroptosis and Its Mechanism

The term “ferroptosis” was first proposed by Dixon et al., in 2012, and it was used to describe a non-apoptotic form of cell death caused by erastin-induced iron-dependent lipid peroxide accumulation [8]. The main morphological characteristics of ferroptosis are the shrinkage of cell volume, the decrease of mitochondrial cristae, and the increase of mitochondrial membrane density without typical apoptotic or necrotic manifestation [8]. The main biochemical changes of ferroptosis are the depletion of glutathione (γ-glutamylcysteinylglycine, GSH) and decrease in glutathione peroxidase 4 (GPX4) activity [8]. Subsequently, the divalent iron ions oxidize lipids in a manner similar to the Fenton reaction, thereby generating a large number of reactive oxygen species (ROS), which in turn promote ferroptosis of cells [29]. GPX4 is the main enzyme that protects the cell membrane against peroxidative damage [30,31], and either its direct or indirect inactivation or an increase in the unstable iron pool will induce the occurrence of ferroptosis [8]. In addition, iron chelator (deferoxamine) and some small molecule compounds (such as ferrostatin-1 (Fer-1) and lipoxstatin-1 (Lip-1)) can reverse the lipid peroxidation caused by ferroptosis [8,29]. It has also been found that nitric oxide synthase (iNOS) may also act as a ferroptosis inhibitor by scavenging lipid free radicals or free radical intermediates [32].

Generally, ferroptosis has a complicated network of genes, proteins, and metabolism. Abnormal iron metabolism and lipid peroxidation may be the main factors that cause ferroptosis, and the system Xc^−^/GSH/GPX4 axis plays a crucial role in this process. In addition, ferroptosis suppressor protein (FSP1), nuclear factor erythroid 2-related factor 2 (Nrf2), p53, and DHODH also have important regulatory effects on ferroptosis. The related molecules and pathways of ferroptosis are shown in Figure 1.

### 2.1. Iron Metabolism and Lipid Peroxidation

Ferroptosis requires iron overload. The iron in the circulation of the human body exists in the form of Fe^3+^. Fe^3+^ is imported into the cell by transferrin and transferrin receptor 1 (TfR1) [7,11] then reduced to Fe^2+^ in the lysosome and, finally, released to the labile iron pool through the divalent metal 9 transporter 1 (DMT1) or zinc iron regulatory protein family 8/14 (ZIP8/14) [33,34]. Excess iron in cells is usually stored in ferritin [35], and ferritin can be recognized by the specific cargo receptor NCOA4, which recruits ferritin to autophagosomes for lysosomal degradation and releases Fe^2+^ [35,36,37,38]. The released Fe^2+^ will generate ROS through the Fenton reaction, and then undergoes a peroxidation reaction with lipids to trigger ferroptosis [29]. Increased iron absorption and reduced iron storage can lead to iron overload. Therefore, inhibiting iron overload by iron chelating agents can inhibit ferroptosis, and silencing the iron metabolism master regulator, iron responsive element binding protein 2 (IREB2), can also reduce the sensitivity of cells to ferroptosis [8].

In addition to abnormal iron metabolism, lipid peroxidation is also an important factor leading to ferroptosis. Lipidomic analysis showed that phosphatidylethanolamines (PEs) containing arachidonic acid (AA) or adrenal acid (AdA) are the key membrane phospholipids, which can be oxidized to phospholipid hydroperoxides (PE-AA/AdA-OOH) through non-enzymatic reactions, thereby driving ferroptosis [39,40]. The main ROS accumulated in cells are superoxide radical anions (·O_2_^−^) and hydrogen peroxide (H_2_O_2_). In the presence of free iron, these ROS may be converted into hydroxyl radicals (HO˙), which are highly reactive to PUFAs that exist in a variety of cell membranes [41]. Free PUFAs, including AA, are oxidized through a catalytic pathway involving acyl-CoA synthetase long-chain family member 4 (ACSL4), lysophosphatidylcholine acyltransferase 3 (LPCAT3), and lipoxygenase (LOXs) [42,43]. ACSL4 and LPCAT3 are key regulators of PUFA-PL biosynthesis [40]. ACSL4 acetylates PUFA to form PUFA-CoA, and then LPCAT3 inserts PUFA-CoA into lysophospholipid to form PUFA-PL [44]. Enzymatic lipid peroxidation is mainly regulated by the LOX family. LOXs can also oxidize PUFAs into corresponding phospholipid hydroperoxides, among which LOX5 and LOX12/15 are involved in ferroptosis [45]. It has been reported that phosphatidylethanolamine binding protein 1 (PEBP1) can form a complex with LOX15 and act as a scaffold protein to positively regulate ferroptosis [46]. It is worth noting here that 15-hydroperoxy (Hp)-arachidonoyl-phosphatidylethanolamine (15-HpETE-PE) produced by the 15-LOX/PEBP1 complex can be eliminated by Ca^2+^-independent phospholipase A2β (iPLA2β) [47]. This indicates that iPLA2β can act as an anti-ferroptotic guardian to regulate the intracellular ferroptosis signaling pathway, which has an important regulatory role in neurodegenerative diseases [47].

In addition, nicotinamide adenine phosphate dinucleotide (NADPH)-dependent lipid peroxidation and the inactivation of GPX4-induced lipid peroxidation are also involved in ferroptosis [48].

### 2.2. The System Xc^−^/GSH/GPX4 Axis

The system Xc^−^/GSH/GPX4 axis is involved in counteracting the endogenous iron–lipid peroxidation-dependent cell death pathway. If any of them are affected, this can trigger the occurrence of cell ferroptosis [49]. The Xc^−^ transporter system consists of a regulatory subunit, solute carrier family 3 member 2 (SLC3A2), and a catalytic subunit, solute carrier family 7 member 11 (SLC7A11), which promote the cellular uptake of cystine by exchange with glutamate [50]. The cystine absorbed into the cell is then reduced to cysteine by GSH or thioredoxin reductase 1 (TrxR1), which in turn can synthesize GSH [51]. GSH is the main antioxidant in mammalian cells and a cofactor of GPX4 [52]. GPX4 can remove lipid hydroperoxides in biological membranes and use GSH to reduce cytotoxic lipid peroxides to non-toxic lipid alcohols, thereby interfering with the lipid peroxidation chain reaction [53]. Therefore, blocking the intracellular cysteine level will directly affect the activity of GPX4 through GSH, thereby increasing the sensitivity of cells to ferroptosis.

### 2.3. Other Related Signaling Metabolism

In addition to the above, other cellular signal pathways also regulate ferroptosis. FSP1 (previously known as AIFM2) is an effective resistance factor. It is independent of GSH and can inhibit ferroptosis by reducing coenzyme Q10 (CoQ10), thereby inhibiting the transmission of lipid peroxides [54,55]. The N-myristoylation of FSP1 is the key to inhibiting ferroptosis, which provides a new target for the development of drugs that target ferroptosis [56].

Nrf2 is a key transcription factor regulating oxidative damage [57]. Under oxidative or electrophilic stress, Nrf2 is released from the Kelch-like ECH-associated protein 1 (Keap1) protein binding and translocates into the nucleus to transcribe the antioxidant response element (ARE)-dependent genes [58,59]. Nrf2 can affect genes encoding GSH synthetic proteins, such as SLC7A11, GCLC/GLCM, and GSS [60]. In addition, Nrf2 can also inhibit ferroptosis by affecting the downstream expression of NQO1, heme oxygenase-1 (HO-1), and ferritin heavy chain (FTH1) [61]. The above process is also regulated by autophagy, and the p62-Keap1-Nrf2 antioxidant signal pathway may be involved in the inhibition of ferroptosis [61].

As one of the most important tumor suppressor molecules, p53 is also involved in the regulation of ferroptosis. It inhibits the system Xc^−^ by down-regulating SLC7A11, thereby inducing ferroptosis [62].

The latest research has found that dihydroorotate dehydrogenase (DHODH) in mitochondria can regulate ferroptosis through a GSH-independent pathway, which provides a new idea for precise targeted regulation of ferroptosis [63].

## 3. Ferroptosis and NAFLD

With the continuous deepening of research, it has been discovered that ferroptosis is closely related to the pathogenesis of many liver diseases. In 2011, researchers discovered iron accumulation in the liver of NAFLD patients [27]. In addition, the iron overload caused by metabolic dysfunction (such as liver ironosis and hereditary hemochromatosis) can aggravate the liver damage of NASH patients [64,65]. Moreover, iron removal can improve liver damage in NAFLD patients [66]. In addition to iron, lipid peroxidation markers (such as malondialdehyde and 4-hydroxynonenal) are also elevated in NAFLD patients [67]. Vitamin E can reduce lipid peroxidation and improve liver damage [68]. These results suggest that ferroptosis caused by iron overload and lipid peroxidation may be involved in the pathological process of NAFLD and that inhibiting ferroptosis may become a new strategy for the treatment of NAFLD.

## 4. Mechanism of Ferroptosis in NAFLD

### 4.1. Decreased GPX4 Activity

Ferroptosis can trigger the inflammatory response of simple fatty liver degeneration, and this promotes the occurrence and development of NASH. In 2019, Tsurusaki et al. found that in the NASH mouse model induced by choline deficiency and ethionine supplementation (CDE) feed, hepatocyte death was accompanied by an increase in the level of oxygenated phospholipid ethanolamine [69]. The researchers also observed that in the initial stage of the development of NAFLD in mice to NASH, hepatocyte ferroptosis precedes cell apoptosis, which in turn leads to liver damage, immune cell infiltration, and inflammation. The ferroptosis inhibitors (e.g., trolox or deferoxamine) can almost completely reverse the death of liver cells, inflammation, and lipid peroxidation in the initial disease model of NASH (Figure 2). It also suppresses the subsequent infiltration of immune cells and inflammatory reaction, thereby improving liver function [69]. This suggests that hepatic ferroptosis, as a trigger of inflammation and initiation of steatohepatitis, may be a therapeutic target to prevent the onset of steatohepatitis.

Qi et al. also found that ferroptosis is a key factor in the development of NASH. Ferroptosis can aggravate the inflammatory response, oxidative stress, and cell damage in the early stages of NASH [70]. GPX4 can inhibit ferroptosis in patients with NASH. Specifically, RSL-3 (GPX4 inhibitor) promoted the progression of NASH by inducing ferroptosis in methionine and choline deficiency (MCD) diet-fed mice (serum biochemical levels and levels of hepatic steatosis and inflammation apoptosis were exacerbated), but sodium selenite (GPX4 activator) improved the severity of NASH [70]. Similarly, in the in vitro NASH model induced by palmitic acid, the regulation of ferroptosis mediated by GPX4 also affects the death of liver cells. Liver is the tissue with the highest expression of GPX4. GPX4 can protect the liver from lipid peroxidation, which is essential for liver function and liver cell survival [71].

In addition, it has been reported that some enzymes in cells can also affect ferroptosis by regulating GPX4. Thymosin β4 (Tβ4) is a multifunctional polypeptide that exists in a variety of nucleated cells. It can improve liver fibrosis and reduce inflammation [72]. Tβ4 can protect hepatocytes by inhibiting the GPX4-mediated ferroptosis pathway [73]. Enolase 3 (ENO3) is another enzyme that encodes the β-subunit of enolase that exists in the liver and in other organs [74]. ENO3 promoted the progression of NASH by negatively regulating ferroptosis via the elevation of GPX4 expression and lipid accumulation [75].

### 4.2. Iron Overload

Iron deposition has been detected in some patients with NASH [66]. Increased iron accumulation due to metabolic damage is a factor that aggravates NASH. Therefore, iron removal therapy can improve liver damage and reduce the level of alanine aminotransferase in the serum [66].

### 4.3. Lipid Peroxidation

The latest research found that in the liver of mice that were fed with MCD feed, the metabolism of AA increased, iron accumulated, lipid ROS enhanced, and the levels of ironophilia-related genes (such as ACSL4, ALOX5AP, GPX4, and PTGS1) increased [76]. It is worth noting that Fe^2+^ is the key factor of ferroptosis, and AA is also the most common PUFA in ferroptosis. Therefore, the increase in Fe^2+^ levels and the increase in AA metabolism synergistically produce lipid peroxidation, which in turn promotes the development of NASH. Treatment of MCD diet mice with Fer-1 and Lip-1 significantly improved liver steatosis, liver damage, inflammation, and fibrosis in the mice, and it reduced the accumulation of lipid droplets and TG [76]. Finally, lipid ROS promotes liver steatosis by promoting the formation of lipid droplets. In terms of mechanism, inhibiting the expression of GPX4 increased the lipid droplets and lipid ROS in cells, while overexpression of GPX4 significantly inhibited the production of lipid droplets. This indicates that GPX4 plays an important role in the homeostasis of lipid droplets [76].

### 4.4. ACSL4 Induction

In addition to GPX4, ACSL4 also plays an important role in the ferroptosis of NAFLD and NASH. Wei et al. also observed ferroptosis in another risk factor of NASH, exposure to arsenic [77]. In both in vitro and in vivo models, exposure to arsenic could up-regulate the expression of ACSL4. However, both the ACSL4 inhibitor rosiglitazone (ROSI) and ACSL4 siRNA can suppress arsenic-induced ferroptosis. In addition, the use of Mitofusin 2 siRNA or IRE1α inhibitor reduced the content of 5-hydroxyeicosatetraenoic acid (5-HETE), which significantly alleviated NASH and ferroptosis [77].

### 4.5. Nrf2 Activation

The Nrf2-mediated antioxidant response plays a key role in the regulation of ferroptosis [78]. Nrf2 can promote the expression of downstream HO-1, GSH, and GPX4, thereby eliminating the accumulation of ROS in the liver and reducing the level of malondialdehyde (MDA) [79]. In addition, activation of the Nrf2 pathway in the obese mouse model can reduce liver lipid accumulation and significantly improve the NAFLD of the mice [80].

### 4.6. Others

A recent study found that enoyl coenzyme A hydratase 1 (ECH1), a key component in mitochondrial fatty acid β-oxidation, can reduce NASH in mice by inhibiting hepatic ferroptosis [81]. ECH1 expression was significantly increased in human NASH biopsy specimens and mice fed with MCD diets. ECH1 overexpression greatly alleviated liver steatosis, inflammation, fibrosis, and oxidative stress. Compared with untreated mice, ECH1-knockdown mice treated with Fer-1 showed a reduction in the NASH phenotype.

The Erk signaling pathway, of course, may be involved, and so this requires further in-depth research.

There is also an article studying the significance of microrna (miRNA) in the pathogenesis of fructose-induced NAFLD, and it found that fructose-induced oxidative damage can induce ferroptosis and that miR-33 can be used as a serological biomarker for fructose-induced NAFLD [82].

Overall, there are relatively few studies on ferroptosis in NAFLD at this stage. The role of ferroptosis in the various stages of NAFLD progression is worthy of further research and exploration, especially since there is no accurate choice for the treatment of NAFLD.

## 5. Drugs Targeting Ferroptosis in Liver Diseases

### 5.1. Ferroptosis Inducers

#### 5.1.1. Erastin and Derivatives

Erastin can promote ferroptosis by inhibiting system Xc^−^ and then causing the depletion of GSH. Piperazine erastin and imidazole ketone erastin are derivatives of erastin [83,84]. Compared with erastin, they have good water solubility and stability. Erastin and its derivatives improve liver fibrosis and hepatocellular carcinoma (HCC) by inducing ferroptosis [61,85].

#### 5.1.2. Sulfasalazine

Sulfasalazine is a first-line anti-inflammatory drug approved by the FDA, and its mechanism of causing ferroptosis is also inhibiting system Xc^−^ [86,87]. In addition, sulfasalazine can also be used to fight cancer by inhibiting the system Xc^−^ of cancer cells [88].

#### 5.1.3. Sorafenib

As a well-known classic drug for the treatment of liver cancer, sorafenib can cause cellular GSH depletion and lipid ROS accumulation by inhibiting system Xc^−^ [89,90,91]. In addition, in response to the clinical resistance of sorafenib to HCC, researchers found that haloperidol (sigma 1 receptor antagonist) can enhance the ferroptosis induced by sorafenib in HCC. This suggests that the combination of haloperidol and sorafenib is a promising treatment strategy for the treatment of HCC [92,93].

#### 5.1.4. Statins

Statins (such as celivastatin, simvastatin, and lovastatin) are inhibitors of HMG-CoA reductase. They reduce CoQ10 by blocking the mevalonate pathway, thereby inducing ferroptosis [94].

#### 5.1.5. Others

Artesunate is a derivative of artemisinin, which can induce hepatic stellate cell (HSC) ferroptosis during liver fibrosis [95]. Mechanically, artesunate activates ferritinophagy (ferritin-targeted autophagy) in liver fibrosis, and this induces ferroptosis [96].

Magnesium isoglycyrrhizinate (MgIG) is also a natural active product with an anti-tumor effect [97]. MgIG can also promote HSC ferroptosis by up-regulating HO-1 in liver fibrosis [98].

### 5.2. Ferroptosis Inhibitors

#### 5.2.1. Fer-1

Fer-1 is a recognized inhibitor of ferroptosis which can reduce PUFAs in membrane oxidation [8]. By inhibiting hepatocyte ferroptosis, Fer-1 improves liver damage mediated by ALD, NAFLD, and hepatic ischemia/reperfusion (I/R) injury [29,76,99] (Table 1). With in-depth exploration of this field, recent research has solved the long-standing Fer-1 anti-ferroptosis paradox. Researchers found that Fer-1 does not affect 15-LOX alone but protects against ferroptosis by effectively inhibiting 15-HpETE-PE produced by the 15-LOX/PEBP1 complex [100].

#### 5.2.2. Vitamin E

Vitamin E is a lipid-soluble radical-trapping antioxidant which can reduce the oxidation level of membrane PUFAs [101]. Studies have shown that a vitamin E diet can make hepatocyte-specific GPX4−/− mice survive normally. This is related to the improvement of hepatocyte degeneration by inhibiting ferroptosis [71]. In addition, vitamin E has also been reported to improve acetaminophen- or I/R-mediated liver damage by inhibiting hepatocyte ferroptosis [102,103].

#### 5.2.3. Dimethyl Fumarate (DMF)

Dimethyl fumarate (DMF) is an activator of Nrf2. It is reported that DMF can reduce lipid peroxidation to inhibit ferroptosis of liver cells, thereby improving ALD [104].

#### 5.2.4. Rosiglitazone (ROSI)

Rosiglitazone (ROSI) is a commonly used thiazolidinedione sensitizer in clinical practice. It can highly selectively activate peroxisome proliferator-activated receptor γ (PPAR-γ), regulate lipid metabolism, and reduce lipid accumulation in liver cells [105]. ROSI can also alleviate ferroptosis by inhibiting ACSL4 [42]. However, ROSI seems to have a potential carcinogenic risk, possibly because its inhibitory effect on ACSL4 reduces the sensitivity of cancer cells to ferroptosis [42].

#### 5.2.5. Others

Dehydroabietic acid (DA) is a kind of natural tricyclic diterpenoid resin acid separated from coniferous plants. It has many benefits to the human body, such as being anti-tumor, anti-bacterial, anti-aging, and anti-inflammatory [106,107,108]. It has been reported that DA improved NAFLD in mice induced by HFD [79]. DA binds to Keap1 and then promotes the expression of HO-1, GSH, and GPX4 downstream of Nrf2, thereby eliminating the accumulation of ROS and reducing MDA. In addition, DA can also increase the expression of key genes such as FSP1 [79].

Ginkgolide B (GB) is the main component of Ginkgo biloba extract. GB inhibits ferroptosis by activating the Nrf2 pathway, which reduces the accumulation of liver lipids in obese mice and improves NAFLD [80].

## 6. Conclusions and Future Directions

Ferroptosis plays an important role in liver pathophysiology, and it also plays a key role in triggering the development of NAFLD to NASH. As a result, it may be a potential therapeutic target to prevent the development of NAFLD. Currently, there are relatively few studies on ferroptosis in NAFLD, but the data clearly suggest that targeted ferroptosis has excellent potential in the prevention and treatment of NAFLD. In view of the gaps in the current research, we believe that the following issues need to be addressed in follow-up research.

### 6.1. Detection of Lipid Peroxide Markers Related to Ferroptosis

In view of the important role that oxidative stress plays in the pathogenesis and severity of NAFLD, many markers of oxidative stress are used to assess the pathological state and progression of NAFLD. Several biomarkers of oxidative stress have been detected in clinical and experimental models of NAFLD, most of which are mainly detected in liver, serum, and plasma [109]. Lipid peroxides that are frequently measured include malondialdehyde, lipid peroxide, 8-isoprostaglandin, and 4-hydroxy-2-nonenal (4-HNE) [110,111]. Therefore, the detection of these markers may have important clinical significance for analyzing whether ferroptosis plays a key role in NAFLD.

### 6.2. The Role of Ferroptosis in Each Stage of NAFLD

It is not known whether ferroptosis plays a role in all stages of NAFLD, including simple steatosis, chronic inflammation, fibrosis, or cirrhosis. These issues need to be further studied.

### 6.3. Clinical Trials and Side Effects of Inhibiting Ferroptosis

The findings discussed above are restricted to preclinical studies. It is still unclear whether the treatment of NAFLD by inhibiting ferroptosis is clinically feasible, and its effectiveness needs to be clinically studied. In addition, as mentioned above, inhibiting ferroptosis may also reduce the sensitivity of cancer cells to drugs. Therefore, future research can consider seeking relevant treatment measures from the multi-target and two-way regulation mechanisms of natural drugs.

### 6.4. The Role of Endotoxin in Regulating Ferroptosis in NAFLD

The role of endotoxin in liver disease has been extensively studied [112], and elevated serum endotoxin levels have been demonstrated in NAFLD [113,114]. In addition, endotoxin has a certain promoting effect on the production of active oxygen intermediate products, such as lipid peroxides, superoxide free radicals, and nitric oxide [115]. Therefore, whether endotoxin can aggravate NAFLD by inducing hepatocyte ferroptosis remains to be further studied.

## Figures and Tables

**Figure 1 biomedicines-09-01660-f001:**
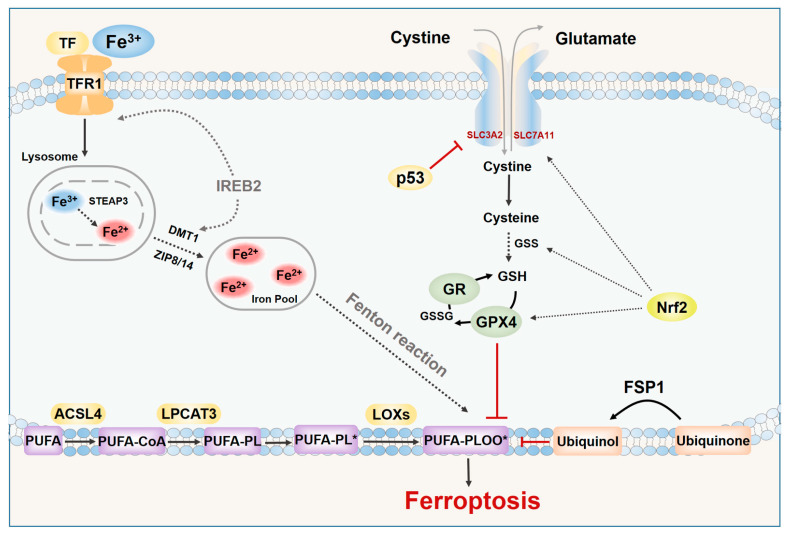
The mechanism of ferroptosis. The figure shows the related molecules and pathways of ferroptosis. Ferroptosis is caused by the inhibition of the system Xc^−^/GSH/GPX4 axis or the accumulation of Fe^2+^ and lipid peroxidation. Regulators of iron metabolism, including TF, TFR1, and ferritinophagy, regulate ferroptosis by enhancing the labile iron pool-mediated Fenton reaction, and FSP1 can negatively regulate ferroptosis through CoQ10, thereby inhibiting the delivery of lipid peroxides. Abbreviations: ACSL4, acyl-CoA synthetase long-chain family member 4; DMT1, divalent metal transporter 1; FSP1, ferroptosis suppressor protein; GPX4, glutathione peroxidase 4; GR, glutathione-disulfide reductase; GSH, glutathione; GSS, GSH synthetase; GSSG, oxidized glutathione; IREB2, iron responsive element binding protein 2; LPCAT3, lysophosphatidylcholine acyltransferase 3; PUFAs, polyunsaturated fatty acids; STEAP3, six transmembrane epithelial antigen of prostate 3; SLC3A2, solute carrier family 3 member 2; SLC7A11, solute carrier family 7 member 11; TF, transferrin; TFR1, transferrin receptor 1; ZIP8/14, zinc/iron regulatory protein family 8/14. PUFA-PL*, polyunsaturated fatty acid-phospholipid radical; PUFA-PL-OO*, polyunsaturated fatty acid-phospholipid ethanolamine peroxyl radical. Black arrows indicate the activation modification, and red block lines indicate inhibitory modification. Black dashed arrows indicate the indirect process. Double arrows indicate the reverse transport process of glutamate and cystein.

**Figure 2 biomedicines-09-01660-f002:**
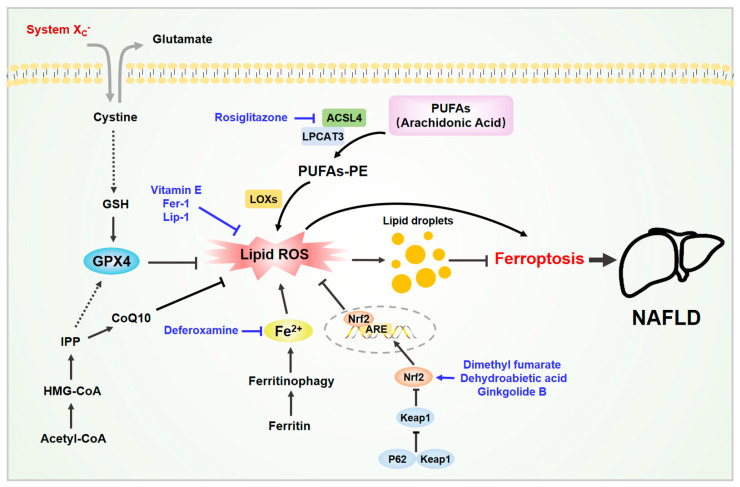
Ferroptosis in non-alcoholic fatty liver disease (NAFLD). Various molecules are involved in ferroptosis in NAFLD. The liver is the main organ for iron storage and lipid metabolism. The accumulation of iron and lipid peroxides in liver cells can induce cell ferroptosis, which in turn exacerbates NAFLD. Some active substances have been shown to inhibit hepatocyte ferroptosis and improve NAFLD by removing excess iron, neutralizing lipid peroxides, and activating the Nrf2 signaling pathway. Abbreviations: ACSL4, acylCoA synthetase long-chain family member 4; CoQ10, coenzyme Q10; GSH, glutathione; GPX4, glutathione peroxidase 4; IPP, isopentenyl pyrophosphate; LPCAT3, lysophosphatidylcholine acyltransferase 3. Black arrows indicate the activation modification, and black block lines indicate the inhibitory modification. Blue arrows indicate the activation process of activators, and blue block lines indicate the inhibition process of inhibitors. Black dashed arrows indicate the indirect process. Double arrows indicate the reverse transport process of glutamate and cystein.

**Table 1 biomedicines-09-01660-t001:** Drugs targeting ferroptosis in liver diseases.

Drugs	Ferroptosis	Target	Mechanism
Erastin [43,44]	Inducer	System Xc^−^	Inhibits system Xc^−^, resulting in GSH depletion
Sulfasalazine [47,48,49]	Inducer	System Xc^−^	Inhibits system Xc^−^, resulting in GSH depletion
Sorafenib [50,51,52]	Inducer	System Xc^−^	Inhibits system Xc^−^, resulting in GSH depletion
Statins [55]	Inducer	HMG-CoA reductase	Depletes CoQ10, resulting in lipid peroxidation
Artesunate [56,57]	Inducer	Ferritinophagy	Activates ferritinophagy
Magnesium isoglycyrrhizinate [59]	Inducer	HO-1	Up-regulates HO-1
Fer-1 [5,11,39,60]	Inhibitor	PUFAs	Inhibits lipid peroxidation
Vitamin E [61,62,63]	Inhibitor	PUFAs	Inhibits lipid peroxidation
Dimethyl fumarate [64]	Inhibitor	Nrf2	Activates Nrf2, leading to reducing lipid peroxidation
Rosiglitazone [19,65]	Inhibitor	ACSL4	Inhibits ACSL4, leading to reducing lipid peroxidation
Dehydroabietic acid [69]	Inhibitor	Nrf2	Activates Nrf2, leading to reducing lipid peroxidation
Ginkgolide B [70]	Inhibitor	Nrf2	Activates Nrf2, leading to reducing lipid peroxidation

Abbreviations: ACSL4, acyl-CoA synthetase long-chain family member 4; CoQ10, coenzyme Q10; GSH, glutathione; HO-1, heme oxygenase-1; Nrf2, nuclear factor erythroid 2-related factor 2; PUFAs, polyunsaturated fatty acids.

## Data Availability

Not applicable.

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
