# Peer review of "Role of Ferroptosis in Non-Alcoholic Fatty Liver Disease and Its Implications for Therapeutic Strategies"

_biomedicines, 2021, doi:10.3390/biomedicines9111660_

Round 1
Reviewer 1 Report
The paper has been significantly improved. The Authors answered correctly to all my questions.
Reviewer 2 Report
Most of the comments were addressed and the quality of the paper has risen.
This manuscript is a resubmission of an earlier submission. The following is a list of the peer review reports and author responses from that submission.
Round 1
Reviewer 1 Report
The manuscript submitted by Han Zhang and co-workers and entitled “Role of ferroptosis in NAFLD and its implications for therapeutic strategies” is a review of the ferroptosis process and its role in non-alcoholic fatty liver disease (NAFLD). The authors described the ferroptosis mechanism, its inducers and inhibitors, and suggest future directions. The manuscript should be significantly improved to be considered for publication. The authors only briefly described the problem and didn’t include some aspects described below. Moreover, in my opinion, the most recent findings (from this year) wasn’t included in the manuscript and previous finding (<2020) were nicely described in review from IJMS 2020 (mentioned below).
To improve the quality of this manuscript I would suggest the following things:
Authors didn’t describe the enzymatic reactions which are proposed in ferroptosis (only brief information, line 84) which should include such proteins as LOX (also in a complex with PEBP1), a first paper about it: https://www.sciencedirect.com/science/article/pii/S0092867417311388 (2017, Cell) and other papers which were published >2017. Therefore, in my opinion, the paragraph which started “Free PUFAs including AA are oxidized through a catalytic pathway..” (line 82) should be improved by adding additional information about the enzymatic hypothesis. It could also provide more information about oxidizable lipids. The authors mentioned that the most common oxidable lipids are AA which is not completely true (even Figure 1 shows PUFA-PLs and Figure 2 PUFAs-PE oxidation of LOXs). Thus, they may want to look at this paper which describes oxidizable lipids studies in the contacts of ferroptosis: https://pubs.acs.org/doi/abs/10.1021/jacs.8b09913.
“In addition, iron chelator (deferoxamine) and some small molecule compounds (such as ferrostatin-1, fer-1 and lipoxstatin-1, lip-1) can reverse the lipid peroxidation caused by ferroptosis[5,11].” – Authors didn’t include the last paper (2021) which explains why ferrostatin blocks ferroptosis process (https://www.sciencedirect.com/science/article/pii/S2213231720309496, with B. Stockwell who discover ferroptosis in 2012). Other recent paper from 2020 show that nitric oxide may be an inhibitor of ferroptosis (https://www.nature.com/articles/s41589-019-0462-8).
The authors didn’t include important findings from 2021, for example:
- Discovery of iPLA2b which averts ferroptosis by eliminating a redox reaction (https://www.nature.com/articles/s41589-020-00734-x) – therefore the statement that “GPX4 is the only enzyme that can remove lipid hydroperoxides” should be modified (line 99)
I am wondering what is the difference between this review and the one from 2020: https://www.mdpi.com/1422-0067/21/14/4908/htm
and why this publication was not cited here: https://www.nature.com/articles/nrgastro.2013.171 (Nature, 2013).
Few references could be added in several places where authors have some important statements, for example:
- It may develop into cirrhosis and liver cancer. (line 22-24)
- GSH is the main antioxidant in mammalian cells and a cofactor of GPX4. (line 96)
The authors included two figures and Table 1 which are not addressed anywhere in the text.
The conclusions don’t bring anything novel except obvious facts.
Reviewer 2 Report
In this paper, Zhang and colleagues review the current knowledge, the future direction and the emerging evidence that link Ferroptosis to Non-alcoholic fatty liver disease (NAFLD). Ferroptosis has been characterized relatively recently as a new pathway that the cell undertakes to die, in response to specific stimuli. Its growing popularity in the scientific community and the continuous increase of papers related to this topic, give an idea of how much this process has been explored in recent years. The possible involvement of this pathway in NAFLD is therefore very interesting and, the work of the authors is noteworthy. Despite this, some concerns arise, especially regarding the cited literature, somehow insufficient, given the development of knowledge in this specific field, and some aspects that must necessarily be deepened and expanded.
Major points:
-In the title no acronyms should be present, and, thus, Non-alcoholic fatty liver disease should be used instead of NAFLD.
-A strong concern regards the literature cited in the paper as it seems a little underestimated. Ferroptosis is now a hot topic in the scientific community, and this is reflected by the works published in these few years. Authors should take into account this properly. For example, as authors state, “Ferroptosis has been reported to play an important role in many human diseases” (line 33), but they only cite a paper that it is not clearly related to this sentence. Ferroptosis has been linked with different pathologies, especially with neurodegerative diseases, including Parkinson’s and Alzheimer’s disease or Friedreich’s Ataxia, for example. Other pathologies have been started to be evaluated and some references are present too. In light of this, multiple references should be given to be more exhaustive, and, probably, the sentence should be expanded a little to be more exhaustive.
-The same occurs for the mechanisms that have been deeply characterized, such as the three main markers of the ferroptotic process (i.e. the GSH-GPX4 axis, the iron overload and the lipid peroxidation). Multiple literature evidence confirmed these hypotheses that permitted to understand how ferroptosis is triggered, and this should be reflected by the papers cited, as these are now cornerstones in many pathological systems and therefore, most likely, fundamental regulators of this pathway.
-Is completely missing that also NRF2 has a strong relationship with Ferroptosis. GPX4 gene is a direct target of NRF2 transcriptional activity, as well as the enzymes responsible of GSH synthesis. This evidence should be included in the text body, in the ferroptosis general mechanism with proper references. In the same way, the relation between NRF2, NAFLD and Ferroptosis could be explored, as reports of NRF2 activity implication in this pathology are present.
- Figures and tables should be cited in the text main body and moved in a proper position (e.g. figure 1 should be at the end of the paragraph describing Ferroptosis mechanisms). Figure 2 legend should be implemented, better describing the figures.
Reviewer 3 Report
The Authors should discuss the emerging role of oxidant stress in relation to lipid peroxidation in NAFLD. They shoul add a paragraph regarding markers of oxidant stress, such as serum sp-NOX2 and urinary 8-iso-PGF1 alpha and the possible relationship with ferroptosis in NAFLD. Finally the possible role of endotoxin, previously related to oxidant stress in NAFLD, in the realization of ferroptosis shoul be analyzed.